# YOLOSAMIC: A Hybrid Approach to Skin Cancer Segmentation with the Segment Anything Model and YOLOv8

**DOI:** 10.3390/diagnostics15040479

**Published:** 2025-02-16

**Authors:** Sevda Gül, Gökçen Cetinel, Bekir Murat Aydin, Devrim Akgün, Rabia Öztaş Kara

**Affiliations:** 1Department of Electronics and Automation, Adapazarı Vocational School, Sakarya University, 54050 Serdivan, Türkiye; gulsevda@sakarya.edu.tr; 2Electrical and Electronics Engineering Department, Engineering Faculty, Sakarya University, 54050 Serdivan, Türkiye; murataydin@sakarya.edu.tr; 3Faculty of Computer and Information Sciences, Software Engineering, Sakarya University, 54050 Serdivan, Türkiye; dakgun@sakarya.edu.tr; 4Department of Dermatology, Sakarya University Training and Research Hospital, 54050 Serdivan, Türkiye; rabiaoztaskara@sakarya.edu.tr

**Keywords:** skin lesion segmentation, object detection, YOLOv8, SAM-Box, artificial intelligence

## Abstract

**Background/Objective**: The rising global incidence of skin cancer emphasizes the urgent need for reliable and accurate diagnostic tools to aid early intervention. This study introduces YOLOSAMIC (YOLO and SAM in Cancer Imaging), a fully automated segmentation framework that integrates YOLOv8 for lesion detection, and the Segment Anything Model (SAM)-Box for precise segmentation. The objective is to develop a reliable segmentation system that handles complex skin lesion characteristics without requiring manual intervention. **Methods**: A hybrid database comprising 3463 public and 765 private dermoscopy images was built to enhance model generalizability. YOLOv8 was employed to localize lesions through bounding box detection, while SAM-Box refined the segmentation process. The model was trained and evaluated under four scenarios to assess its robustness. Additionally, an ablation study examined the impact of grayscale conversion, image blur, and model pruning on segmentation performance. **Results**: YOLOSAMIC demonstrated high segmentation accuracy, achieving Dice and Jaccard scores of 0.9399 and 0.9112 on the public database and 0.8990 and 0.8445 on the hybrid dataset. **Conclusions**: The proposed YOLOSAMIC framework provides a robust, fully automated solution for skin lesion segmentation, eliminating the need for manual annotation. Integrating YOLOv8 and SAM-Box enhances segmentation precision, making it a valuable decision-support tool for dermatologists.

## 1. Introduction

Skin cancer is the fifth most commonly reported cancer in the world as of 2020, according to the World Health Organization, with a total of 1.20 million cases of skin cancer, excluding melanoma [1]. Skin cancers are usually classified into two main types: melanoma, which originates from melanocytes, and nonmelanoma skin cancer, which arises from keratinocytes. Nonmelanoma skin cancer is diagnosed depending on its severity as basal cell carcinoma (BCC) or squamous cell carcinoma (SCC) [2]. Globally, 2 to 3 million nonmelanoma skin cancers and 132 thousand melanoma skin cancers are reported to occur each year [3]. Although 75% of reported skin cancer cases are BCC and 20% are SCC, invasive melanoma is responsible for 80% of skin cancer-related deaths [4,5].

Early diagnosis, timely intervention, and follow-up screening programs are essential to reduce mortality, morbidity, and treatment costs. However, manual diagnosis is time-consuming and requires the opinion of a dermatologist. The lack of a sufficient number of dermatologists and challenging lesion characteristics make early diagnosis difficult. In the absence of dermatologists, patients are examined by general practitioners, and their diagnostic accuracy varies between 24 and 70%. On the other hand, the agreement between dermatologists and practitioners is around 57% [6,7]. As a result, computer-aided diagnostic and decision-support systems have gained importance.

### Literature Review

Recent advancements in image processing, machine learning, and artificial intelligence techniques provide promising results in early skin cancer detection. Identifying the skin lesion area is one of the fundamental steps in designing a helpful decision support system for experts. For this purpose, various segmentation methods have been proposed in recent years [8,9]. The recent and valuable studies performed to improve the accuracy and efficiency of automated skin lesion segmentation were investigated to show the contributions of the proposed research.

The study in [10] used the International Skin Imaging Collaboration (ISIC) 2019 database, which is divided into two main sets: training and testing. The training set consists of 1440 benign and 1197 malignant images, while the test set contains 360 benign and 300 malignant images. Each image in the database was standardized to size 224 × 224. A skin cancer detection method was introduced using the proposed Fractional Student Psychology Based Optimization-based Deep Q Network (FSPBO-based DQN) to detect skin lesions early. The accuracy, sensitivity, and specificity percentages were 92.364%, 93.20%, and 92.63%, respectively [10].

The study in [11] introduced a hyper-parameter optimized Fully Convolutional Encoder–Decoder Network (FCEDN) for dermoscopy image segmentation. The novel Exponential Neighborhood Grey Wolf Optimization (EN-GWO) algorithm was employed to optimize network hyper-parameters. EN-GWO incorporates a neighborhood-based searching strategy, combining individual and global haunting strategies of wolves to strike a balance between exploration and exploitation. The study compared EN-GWO with four variants of Grey Wolf Optimization (GWO), Genetic Algorithm (GA), and Particle Swarm Optimization (PSO) for hyperparameter optimization on the ISIC 2016 and ISIC 2017 databases. The proposed model achieved a Jaccard index of 96.41% and 86.85%, a Dice coefficient of 98.48% and 87.23%, and an accuracy of 98.32% and 95.25% for the ISIC 2016 and ISIC 2017 databases.

Mingzhe and his friends introduced SkinSAM, a finely tuned model derived from the SAM, showcasing remarkable segmentation performance. The evaluation was conducted on the HAM10000 database, encompassing 10,015 dermoscopy images. At the same time, larger models outperformed the smaller ones; the fine-tuned model demonstrated the most significant enhancement, achieving a mean pixel accuracy of 0.945, a mean dice score of 0.8879, and a mean Intersection over Union (mIoU) score of 0.7843. Vascular lesions exhibited the most precise segmentation results among the various lesion types [12].

The study managed by Akyel et al. combined two databases from ISIC and PH2 to explore a novel approach (LinkNet-B7) for denoising and segmenting skin cancer images. LinkNet-B7, an adaptation of the LinkNet architecture incorporating EfficientNetB7 as the encoder, demonstrated a 6% improvement in success rates compared to LinkNet with identical databases and parameters. The training accuracy achieved for noise removal and lesion segmentation was notably high, reaching 95.72% and 97.80%, respectively [13].

Then, the paper in [14] suggested a novel approach named Multiscale Attention U-Net (MSAU-Net), designed for skin lesion segmentation. The enhancement involved integrating an attention mechanism into the typical U-Net architecture at the network’s bottleneck, effectively capturing hierarchical representations. The attention module orchestrated a non-linear aggregation of multi-level representations, allowing for selective adjustment of representative features. Additionally, the model utilized a Bidirectional Convolutional Long Short-Term Memory (BDC-LSTM) structure to emphasize standard discriminative features while suppressing less informative ones. The resulting features were integrated into each decoding path block to emphasize crucial regions. The proposed network was thoroughly evaluated on three public skin lesion databases, ISIC 2017, ISIC 2018, and PH2, to prove its efficacy in segmentation tasks. The Jaccard values achieved were 0.9576, 0.956, and 0.9617 for ISIC 2017, ISIC 2018, and PH2 databases, respectively.

Another study proposed a melanoma segmentation approach, including U-net and LinkNet deep learning networks, coupled with transfer learning and fine-tuning techniques. Experiments conducted on three publicly available databases (PH2, ISIC 2018, and DermIS) have shown promising results, with U-net demonstrating notable performance. Specifically, the average Dice coefficient achieved was 0.923 on the PH2 database, 0.893 on ISIC 2018, and 0.879 on the DermIS database. These findings indicate significant success for U-net across the evaluated databases [9].

A pioneering Multi-class Dilated D-Net (MD2N) framework for segmenting and classifying diverse skin cancer types was given in [15]. In the encoder phase of MD2N, a downsampling ratio was employed to mitigate feature information losses and effectively discern small skin lesion patches. Consequently, the model captured richer feature information from skin lesion areas of varying sizes. Evaluation results demonstrated the model’s performance, achieving an accuracy of 97.462%, precision of 93.627%, recall of 99.721%, F1 score of 97.182%, and specificity of 93.289%.

Reis et al. proposed InSiNet, a convolutional neural network based on deep learning for detecting benign and malignant skin lesions. Under consistent conditions, the method’s performance was assessed using images from the ISIC 2018, ISIC 2019, and ISIC 2020 databases. Comparative computation time and accuracy analysis were conducted between InSiNet and various machine learning techniques, including GoogleNet, DenseNet-201, ResNet152v2, Efficient-NetB0, RBF-support vector machine, logistic regression, and random forest. The results demonstrated that InSiNet achieves accuracy rates of 94.59%, 91.89%, and 90.54% on the ISIC 2018, ISIC 2019, and ISIC 2020 databases, respectively [16].

An innovative Convolutional Neural Network (CNN) framework utilizing atrous convolutions for automatic lesion segmentation is given in [17]. The architecture was constructed based on the concept of atrous/dilated convolutions, which are known for their effectiveness in semantic segmentation. The deep neural network was built, beginning with a comprehensive design approach. It incorporated fundamental elements such as convolutional layers, batch normalization, leaky ReLU layers, and adjusted hyperparameters to optimize performance. The network underwent testing on three benchmark databases: ISIC 2016, ISIC 2017, and ISIC 2018. The proposed network achieved an average Jaccard index of 90.4% on ISIC 2016, 81.8% on ISIC 2017, and 89.1% on ISIC 2018 databases, respectively.

The CNNs and the integration of handcrafted features have shown promising results in the automated detection of skin lesions. The SNC-Net framework, which merges handcrafted image features with deep learning models to improve skin cancer detection, is proposed in [18]. This hybrid approach has significantly improved lesion classification, achieving higher precision and recall rates in distinguishing between benign and malignant skin lesions. The ISIC 2019 database achieved state-of-the-art performance with an accuracy of 97.81%, a precision of 98.31%, a recall of 97.89%, and an F1 score of 98.10%. The proposed model outperformed EfficientNetB0, MobileNetV2, DenseNet-121, and ResNet-101 baseline models and state-of-the-art classifiers. Furthermore, the method’s robustness was validated through an ablation study [18].

Iqbal et al. developed a deep convolutional neural network for multi-class classification of skin lesions, demonstrating a remarkable balance between efficiency and accuracy [19]. In another study, the authors explored the application of transfer learning in medical imaging by detecting synovial fluid in knee MRIs, highlighting the adaptability of deep learning techniques to different medical domains [20]. These studies’ efforts in integrating CNNs with handcrafted features emphasize the importance of robust model architecture for accurate and efficient lesion segmentation.

The systematic review in [21] analyzed 164 studies on deep learning applications for skin cancer detection, highlighting AlexNet, ResNet-50, VGG-16, and GoogLeNet as the leading architectures for achieving optimal classification results. Multiclassification techniques are identified as a growing trend, and the availability of robust public databases is emphasized as essential for advancing research in this field. Then, another recent systematic literature review examines federated learning and transfer learning techniques widely applied in skin cancer classification, analyzing their performance based on metrics such as true positive rate, true negative rate, area under the curve, and accuracy. The study reviewed 86 articles published in reputable forums between January 2018 and July 2023, utilizing data from seven prominent databases. A taxonomy summarizing malignant and non-malignant skin cancer classes is also presented. The review identifies key limitations and challenges in existing research, providing insights into future opportunities for automated melanoma and nonmelanoma classification, thereby guiding researchers in advancing this critical healthcare area [22].

Thus, the main contributions of the proposed study can be emphasized as follows:Hybrid Integration of Object Detection and Segmentation: Unlike conventional approaches that rely solely on segmentation networks, the proposed study combines YOLOv8’s precise lesion localization with SAM-Box’s advanced segmentation capabilities, achieving fully automated lesion extraction without manual intervention.Extensive Hybrid Database for Clinical Generalizability: The constructed hybrid dataset introduces real-world challenges, including lesions in diverse skin tones, images with hair occlusions, and complex backgrounds to enhance model generalization beyond public databases.Evaluation Across Multiple Training Strategies: The system was tested under four distinct training/testing scenarios. The results demonstrate superior generalization when incorporating hybrid databases, providing insights into optimizing model training strategies for clinical applications.State-of-the-Art Performance in Automated Skin Lesion Segmentation: The proposed approach achieves exceptional segmentation accuracy, with Dice and Jaccard indices of 0.9399 and 0.9112 on public databases and 0.8990 and 0.8445 on hybrid databases. These results ensure the system’s real-time inference capability.Comprehensive Ablation Study for Performance Analysis: The presented study conducts a detailed ablation analysis, evaluating the impact of grayscale images, image blur, and model pruning on segmentation performance. The ablation study provides critical insights into the robustness and computational efficiency of YOLOv8 under different conditions.Benchmarking YOLOv8 Against YOLOv11 for Computational Efficiency: The performances of YOLOv8 and YOLOv11 were evaluated regarding segmentation accuracy, computational efficiency, and real-time applicability. The results indicate that YOLOv8 achieves higher segmentation accuracy with lower computational overhead. This balance between accuracy and efficiency makes YOLOv8 a more suitable candidate for real-time clinical applications.

## 2. Materials and Methods

In this section, the constructed database was first introduced. Then, the segmentation process and the algorithms utilized were explained to help the study’s methodology be understood better.

### 2.1. Comprehensive Database

The extended database of the study was built by combining public and private dermoscopy images. A total of 4228 images were utilized in the study, comprising 3463 images acquired from public databases and 765 images sourced from private databases. The comprehensive database encompasses diverse dermoscopy images corresponding to various lesion types. The database was designed to include lesion images representing real-world challenges to ensure a thorough and reliable evaluation of the proposed segmentation process. These lesions are in individuals with dark skin tones, where the similarity between the lesion and the surrounding skin makes detection difficult, as well as lesions partially obscured by hairs. The database also features small lesions, images with complex backgrounds, lesions that span the entire imaging field, and those closely matching the skin tone. By integrating these varied scenarios, the segmentation method is rigorously tested against different realistic conditions to evaluate its capabilities.

The public portion of the database was obtained from ISIC, which focuses on improving the early detection and diagnosis of skin cancer through digital imaging. This collaborative project engages researchers, clinicians, and industry experts to create and share a comprehensive skin image database for research and education. The annual ISIC challenges serve as competitive platforms, inviting researchers and data scientists to devise algorithms facilitating the automated analysis of skin lesions. The ISIC challenges from 2016 to 2018 contributed to establishing the study’s public database [23].

In the context of the segmentation task, 7723 images were sourced from the ISIC databases from 2016 to 2018. The presented study removed duplicated images from the combined ISIC databases, resulting in 4075 unique images. Notably, due to the absence of liquid usage during image acquisition in the private database (which was the primary focus of the developed segmentation process), images with bubbles were systematically excluded from the ISIC databases. Consequently, a public database containing 3463 images and ground truth masks was compiled. The private database for the study, which included 765 images (752 h × 582 w), was constructed by a dermatologist from Sakarya University Training and Research Hospital with the permission of the ethics committee. The dermatologist manually determined the lesion area for the images in the private database, thus creating ground truth masks.

Fifteen benign and malignant skin lesion types included melanoma, melanocytic nevus, lentigo, seborrheic keratosis, angioma, actinic keratosis, Bowen’s disease, trichilemmal cyst, keratoacanthoma, hemangioma, malignant melanoma, basal cell carcinoma, pyogenic granuloma, Spitz nevus, and Merkel cell carcinoma. The training, validation, and test sets were assembled by integrating images from public and private databases. These databases were systematically divided into proportions of 80%, 10%, and 10%, respectively. The database preparation process is illustrated in Figure 1.

The hybrid database in this study combines public and private dermoscopic images to improve model generalizability and reflect real-world clinical challenges. However, private clinical images introduce additional complexities that can impact segmentation performance. These images exhibit more significant variability in resolution, illumination conditions, and background artifacts compared to standardized public databases. Additionally, the private database includes a broader range of lesion types, including rare and atypical cases, which may not be adequately represented in public databases, potentially affecting the model’s generalization ability. Another important factor is skin tone diversity, as the private database encompasses a broader spectrum of skin tones, influencing image contrast and segmentation accuracy. These factors contribute to the observed performance differences when training on public versus hybrid databases. To further enhance the robustness of the YOLOv8 and SAM-Box system, ongoing efforts are focused on expanding the dataset with additional clinical images from diverse sources, ensuring broader applicability across different patient demographics and clinical settings.

### 2.2. YOLOv8 Architecture

Real-time object detection is necessary in many systems, from autonomous vehicles to medical devices. Although various algorithms have been developed for object detection, researchers prefer YOLO due to its advantages, especially in speed and accuracy. In addition, while YOLO gives the user the coordinates of the bounding box where the object is located, it can also undertake the classification task. In response to its interest, YOLO architecture has continued to improve since the day it was first developed. YOLO architecture is used in medical applications such as lesion segmentation, tumor detection, and pill identification to support specialists in deciding the most effective treatment processes [24].

YOLOv8 was introduced in 2023 by the Ultralytics company, which is also the developer of YOLOv5. The YOLOv8 model comes in alternative configurations: nano, small, medium, large, and x-large. Each configuration is designed to meet different computational resources and performance needs to offer options for efficient and accurate object detection tasks. In Figure 2, YOLOv8 architecture was demonstrated. The input layer of the model uses the sigmoid activation for the objectness score. The backbone of YOLOv8 is like that of YOLOv5, with some modifications on the cross-stage partial connections (CSP) layer referred to as the C2f module, using high-level features and contextual knowledge to strengthen detection ability. The backbone of the YOLOv8 is the CSPDarknet53 feature extractor. A spatial pyramid pooling fast (SPPF) layer is included in the network to speed up the computation by pooling features into a fixed-size map. Batch normalization and SiLU activation are carried out in each convolution block. The head layer is connected to the outputs of the three C2f modules and performs object detection, classification, and regression tasks independently. YOLOv8 is developed using the Pytorch framework, providing 53.9% average precision (AP) on the COCO2017 dataset.

The architectural components and mathematical formulations employed in YOLOv8, including the backbone, neck, and head networks, are summarized below.

*Backbone Network*: The backbone network in YOLOv8 is responsible for feature extraction from the input images. It utilizes a modified CSPDarknet architecture, which includes the following components:Convolutional layers apply convolution operations to extract fundamental image features such as edges and textures. The convolution operation is defined as(1)$$y=f\left(W*x+b\right)$$
where *y* represents the output feature map, *x* is the input feature map, *W* denotes the convolution kernel, *b* is the bias term, and *f* is a non-linear activation function, typically the Rectified Linear Unit (ReLU).
2.Residual blocks facilitate the training of deeper networks by enabling the gradient to flow more effectively through the network. A residual block can be mathematically expressed as (2)$$y=f\left(x\right)+x$$
where *f*(*x*) denotes the output of the convolutional layers within the block.

3.CSP layers divide the feature map into two parts and merge them through a cross-stage hierarchy. This design enhances gradient flow and reduces computational complexity.

*Neck Network*: The neck network in YOLOv8 is designed to enhance the feature maps for object detection at multiple scales. It incorporates the following two networks:The Feature Pyramid Network (FPN) structure allows for multiscale feature extraction by combining low-resolution, semantically rich features with high-resolution, semantically weak features. The merging of features at different scales is mathematically expressed as(3)$$P_{i}=F_{i}+UpSample(P_{i+1})$$
where *P*_*i*_ represents the feature map at level *i*, and *F*_*i*_ is the feature map from the backbone network.
2.The Path Aggregation Network (PANet) improves information flow by incorporating a bottom-up path aggregation. The PANet equation can be given as follows:(4)$$P_{i}^{\prime }=F_{i}+DownSample(P_{i-1}^{\prime })$$
where *P*_*i*_′ denotes the enhanced feature map at level *i*.

*Head Network:* The head network in YOLOv8 generates the final predictions, including the class labels, bounding box coordinates, and confidence scores. YOLOv8 employs an anchor-free detection head architecture.
Unlike the previous YOLO versions that use predefined anchor boxes in anchor-free prediction, YOLOv8 directly predicts object centers. The bounding box prediction involves the regression of offsets relative to grid cell centers: (5)$$t_{x}=\sigma \left(x\right)+c_{x}$$
(6)$$t_{y}=\sigma \left(y\right)+c_{y}$$
(7)$$t_{w}=p_{w}e^{w}$$
(8)$$t_{h}=p_{h}e^{h}$$
where *t*_*x*_ and *t*_*y*_ are the predicted center coordinates, *t*_*w*_ and *t*_ℎ_ are the predicted width and height, *c*_*x*_ and *c*_*y*_ are the grid cell coordinates, and *p*_*w*_ and *p*_ℎ_ are the anchor box dimensions.
2.The network predicts a confidence score indicating the presence of an object and class probabilities for each bounding box using a sigmoid activation function: (9)confidence=σ(c)
(10)class probabilities=Softmax(p)

where *c* is the raw confidence score, and *p* represents the raw class score logits.

The loss function in YOLOv8 combines localization loss, confidence loss, and classification loss to optimize the network. It is expressed as(11)$$L=\lambda_{coord}L_{coord}+L_{conf}+\lambda_{class}L_{class}$$
where $L_{coord}$ is the localization loss (e.g., smooth *L*_1_ loss for bounding box regression), $L_{conf}$ is the confidence loss (e.g., binary cross-entropy), and $L_{class}$ is the classification loss (e.g., categorical cross-entropy). The hyperparameters $\lambda_{coord}$ and $\lambda_{class}$ balance the contributions of each loss component. YOLOv8 is trained using a large-scale dataset with annotated images to ensure the detection of various objects with high accuracy. The training process involves optimizing the network parameters to minimize the combined loss function. Evaluation is performed on a separate validation dataset to assess the model’s precision, recall, and mean Average Precision (mAP) performance.

In this study, YOLOv8 (Large) is used to determine the coordinates of the bounding box outlining the area with the lesion. Then, the obtained box coordinates are applied to the SAM-Box input.

### 2.3. Segment Anything Model

The SAM, introduced in 2023, represents a significant leap in image segmentation. Built upon a robust foundation of masked autoencoders (MAE) and vision transformers (ViT), SAM has been trained on the largest segmentation dataset, SA-1B, which includes over 1 billion masks and 11 million images. The core aim behind SAM’s design is to provide a universal segmentation solution capable of performing well across a wide array of image types without requiring retraining or fine-tuning for specific tasks. This flexibility makes SAM a valuable tool for various image segmentation applications, including those in complex fields like medical imaging [25]. SAM operates through three key components, each playing a critical role in its segmentation process:

*Image Encoder*: The image encoder extracts meaningful features from input images. Using a ViT, the model divides an image into fixed-size patches, processes them through transformer layers, and captures intricate spatial relationships. ViT’s architecture excels in handling large and high-resolution images, making it particularly effective for tasks like segmentation, where fine details and precise boundaries are crucial.

SAM offers three versions of ViT architecture, each tailored for different levels of complexity. ViT-B is the base version, designed for general use and optimized for computational efficiency while maintaining strong performance. Conversely, ViT-L is a larger version with more parameters, designed to handle more complex tasks requiring deeper feature extraction. ViT-H is the most advanced version, offering the highest level of accuracy, particularly in tasks where precision is critical.

*Flexible Prompt Encoder*: One of SAM’s distinguishing features is its ability to adapt to various input prompts, which guide the segmentation process. These prompts can be sparse, such as a point, box, or text, or dense, like masks that provide pixel-level annotations. The prompt encoder processes these different prompt types using self-attention mechanisms, enabling SAM to adjust its focus based on user input. This flexibility benefits users who wish to guide the model’s attention to specific regions of interest, enhancing segmentation accuracy.

*Fast Mask Decoder*: The final component in SAM is the fast mask decoder, which generates segmentation masks based on the features extracted by the encoder. This decoder combines cross-attention and self-attention to refine the mask predictions. Cross-attention allows the model to align features from the image and the provided prompt, while self-attention enables the decoder to update and improve its predictions iteratively. This attention-driven refinement process ensures high-quality segmentation outputs, even in complex objects or boundaries requiring precise delineation. Figure 3 gives the SAM overview as used in the proposed study.

Since the introduction of SAM, its performance has been investigated in various medical applications. It is still being analyzed whether the SAM can be applied to more challenging medical image segmentation tasks due to the structural complexity, low contrast, and inter-observer variability of medical images. For this purpose, studies summarized under the following subheadings have been carried out: (i) pathology image segmentation, (ii) liver tumor segmentation from contrast-enhanced computed tomography (CECT), (iii) polyps segmentation from colonoscopy images, (iv) brain magnetic resonance imaging (MRI) segmentation, (v) abdominal computed tomography (CT) organ segmentation, (vi) endoscopic surgical instrument segmentation, and (vii) other mixed dataset evaluations. In these segmentation studies, three variations of SAM (SAM-Semantic, SAM-Point, and SAM-Box) were used, and evaluations were made by calculating a standard metric referred to as the Dice coefficient. According to the results of the studies, SAM provides significantly lower Dice values than existing deep learning-based methods used for medical image segmentation, such as U-Net architectures [26,27,28,29,30,31].

On the other hand, SAM-Box and SAM-Point methods slightly increase the segmentation performance compared to SAM-Semantic because they utilize user-provided prior information for the possible object region. However, requiring user interaction is an essential limitation of SAM-Box and SAM-Point methods in fully automatic segmentation applications. According to the parameter sizes, SAM can be used with three vision transformers: ViT-B, ViT-L, and ViT-H. In this study, the experimental results were achieved by performing all versions of SAM. The ViT-H model was selected as it provided the best performance regarding segmentation accuracy.

After introducing the database and the methods utilized, the flowchart of the proposed system can be summarized to better illustrate the study’s contributions. Figure 4 shows the primary stages of the proposed skin lesion segmentation system.

The main target of the study was to develop a fully automatic system that performs skin lesion segmentation tasks without intervention. As can be seen from the figure, the proposed skin lesion segmentation system consists of three main processes: (i) data preparation, (ii) box detection, and (iii) segmentation and evaluation.

The first step of the data preparation process was to build a comprehensive database consisting of dermoscopy images taken from various skin lesion types. The study database includes public and private lesion images, as illustrated in Figure 1. During the labeling process, only “lesion” regions were marked. This decision was made to focus on segmentation and enable the model to identify lesions accurately. By annotating solely the lesion areas, the aim was to streamline the segmentation task and enhance the model’s ability to detect lesions precisely. Then, the labeled images were used to train, validate, and test the YOLOv8 (Large). Before applying the data augmentation methods to the training images, all database images were resized to 256 × 256. The number of images in the training set has quadrupled through augmentation methods, which include vertical/horizontal flipping, brightness adjustment within the [−25% to +25%] range, and rotation within the [−15° to +15°] range.

During the box detection process, four training/test scenarios were created to evaluate the performance of object detection and segmentation methods reliably. The mAP, precision, and recall metrics were calculated to assess the performance of the YOLOv8 (Large) detector for each scenario. The final step of the system involved the segmentation and evaluation process, which aimed to determine the skin lesion area using SAM-Box automatically. In this step, the bounding box coordinates provided by the YOLOv8 (Large) were inputted into SAM-Box, and skin lesion masks were generated without any manual intervention. Finally, the skin lesion segmentation performance of the entire system was evaluated using five standard metrics by comparing the automatically generated masks with the ground truth masks.

## 3. Results and Discussions

The study aims to perform skin lesion segmentation without any intervention. The most up-to-date techniques have been used to achieve this goal. The study sequentially conducted data preparation, box detection, segmentation, and evaluation processes.

The performance analysis of the proposed system has been examined under two main headings. The efficacy of the YOLOv8 object detection algorithm was assessed in the first subsection. In contrast, the second subsection focused on the success of the SAM-Box employed for lesion segmentation. All simulations were conducted using Python 3.10.9.

### 3.1. Object Detection Results

The effectiveness of the YOLOv8 was initially evaluated using commonly employed metrics, followed by an analysis of loss functions to assess its performance. At first, precision, recall, and mAP metrics were considered to evaluate YOLOv8’s performance.

Precision is a fundamental metric in evaluating object detection algorithms and measuring the accuracy of positive predictions. It quantifies the ratio of true positive predictions to the sum of true positive and false positive predictions. Precision is crucial in applications where minimizing false alarms is paramount, such as in security and surveillance systems. Recall, synonymous with sensitivity, assesses the ability of an object detection algorithm to identify all instances of a target object. It is calculated as the ratio of true positive predictions to the sum of true positive and false negative predictions. Recall is of utmost importance in applications where missing object instances can have severe consequences, as in the case of autonomous vehicles.

The mAP score, being a single scalar value, facilitates direct model comparisons and benchmarking. The mAP stands out as a comprehensive metric in object detection evaluation. It considers precision at multiple recall levels, providing a nuanced understanding of an algorithm’s performance. The mAP score is obtained by calculating the average precision across all object classes, offering a unified measure of detection accuracy. A higher mAP score indicates superior performance, making it a widely adopted metric in benchmarking object detection models.

These metrics offer a comprehensive view of an algorithm’s strengths and weaknesses, guiding researchers and practitioners in making informed decisions regarding algorithm selection and deployment. As object detection technology continues to advance, the refinement and standardization of evaluation metrics will play a crucial role in ensuring the reliability and scalability of these algorithms across diverse applications. Furthermore, object detection models often comprise three main components: objectness prediction (object or not), bounding box regression (box coordinates), and class prediction (classification of the object). Each of these components has its associated loss function. The object loss focuses on determining whether an object is present, the box loss refines the coordinates of the bounding box, and the class loss handles the classification of the object. Combining these three components and their respective loss functions contributes to training a comprehensive object detection model capable of accurately localizing and classifying objects in an image.

As discussed before, the YOLOv8 (Large) model was used to perform the object detection task of the proposed study. According to the literature review, open databases were used in a significant part of the studies conducted for skin lesion segmentation. In the presented study, a private database was created, and a hybrid database was obtained by combining it with public databases. So, the performance of the object recognition algorithm was analyzed in two separate cases during the simulations. In the first case, training was conducted on public databases to compare the presented study with previous studies. In the second case, training was carried out on the hybrid database to assess the system’s performance in clinical applications.

The training performance of YOLOv8 was investigated concerning the loss functions, precision, recall, and mAP metrics. Figure 5 and Figure 6 show the performance graphics for “training on public databases” and “training on hybrid database” cases. According to the figures, the loss values vanish to zero as the number of epochs increases. The convergence of class and box loss toward zero collectively enhances the object detection algorithm’s ability to accurately classify objects, precisely predicting bounding box coordinates. This convergence is indicative of a more refined and effective object detection model. The dual focal loss (DFL) curve was also given for YOLOv8. DFL considerably improves object detection by addressing class imbalance and handling complex examples more effectively than traditional focal loss. DFL evaluates the model’s robustness to real-world challenges and improves its generalization capabilities. A diminishing DFL indicates that the model is becoming increasingly proficient at assigning appropriate emphasis to different examples, resulting in improved performance on challenging instances in the database. As can be seen from the figures, the algorithm training process was completed with high performance.

Figure 7 illustrates the lesion detection (test) performance of YOLOv8 for dermoscopy images, including various challenging cases. Specifically, the evaluation considers lesions in individuals with dark skin tones and those partially obscured by hairs. Furthermore, the method’s performance is investigated on small lesions, images with complex backgrounds, and lesions that span the entire imaging field. Interpreting segmentation results of challenging cases is crucial to reliably evaluating the performance of the proposed system.

In Figure 7, the columns were bounding boxes obtained by YOLOv8, segmentation masks provided by SAM-Box, binary segmentation masks, ground-truth masks, and difference images of predicted output and ground truth images, respectively. Note that in cases where multiple bounding boxes are detected, the algorithm proceeds with the box having the highest computed similarity score. As can be seen from Figure 7, the segmentation results of the utilized methods meet clinical expectations.

### 3.2. Segmentation Results

The last process of the presented study is segmentation and evaluation. Accurate segmentation is imperative for biomedical image processing applications, and comparing manual and automatic segmentation results is essential for gauging the reliability of automated algorithms. The proposed study aims to segment skin lesions with the highest accuracy without intervention.

Various metrics were calculated by comparing the predicted masks with the ground truth masks to evaluate the performance of algorithms used for segmenting skin lesions. The Dice coefficient and Intersection over Union (IoU) are two prevalent similarity measures. Dice is calculated as twice the overlap area between the predicted and ground truth masks, divided by the total area of both masks, yielding a value between 0 and 1, with 1 indicating perfect overlap. IoU or Jaccard index, on the other hand, is the ratio of the intersection area to the union area of the predicted and ground truth masks. These metrics are crucial for assessing the model’s performance, especially in datasets with limited examples. The equations for Dice and IoU metrics can be given as follows:(12)$$Dice=\frac{2\left|A\cap B\right|}{\left|A\right|+\left|B\right|}$$(13)$$IoU=\frac{\left|A\cap B\right|}{\left|A\cup B\right|}$$

$\left|A\right|$ and $\left|B\right|$ are the number of pixels in the predicted and ground truth segmentation sets. $\left|A\cap B\right|$ and $\left|A\cup B\right|$ show the intersection and union of these sets, respectively.

On the other side, the pixel-level confusion matrix, which includes true positive (*TP*), false positive (*FP*), true negative (*TN*), and false negative (*FN*) values, provides a detailed breakdown of the model’s predictions. Accuracy, calculated as the ratio of correctly predicted pixels to the total number of pixels, offers an overall measure of the model’s correctness. Precision, the proportion of true positive predictions among all positive predictions, indicates the model’s ability to avoid false positives. Recall that the proportion of true positive predictions among all actual positives reflects the capacity of the model to detect all relevant cases. These metrics comprehensively evaluate the model’s strengths and weaknesses, enabling precise adjustments to enhance performance. The equations of the metrics can be expressed as follows:(14)$$Precision=\frac{TP}{TP+FP}$$(15)$$Recall=\frac{TP}{TP+FN}$$(16)$$Accuracy=\frac{TP+FP}{TP+TN+FP+FN}$$

In the proposed study, SAM-Box was used for skin lesion segmentation purposes. However, as previously mentioned, the SAM-Box model requires box coordinates to be provided as a prompt. While open-source SAM is widely utilized for segmentation in many applications, prompts are often given manually, and SAM-Semantics, which does not require a prompt, is preferred. When prompts are provided manually, the system loses its automatic functionality, but using SAM-Semantics results in multiple candidate regions from the segmentation. This study evaluated the YOLOv8 (Large) model to obtain region box coordinates automatically.

Figure 8 illustrates the segmentation process applied to the original image through two approaches. The results of applying SAM-Semantics directly to the image are shown in the upper arm. In the lower arm, the outputs are obtained by applying YOLOv8 for object detection, followed by SAM-Box for segmentation, as suggested in the study. This comparison highlights the differences between using SAM-Semantics alone and combining YOLOv8 with SAM-Box for image segmentation. The ground truth mask is provided at the end of the figure to validate the performance of the proposed segmentation system.

The figure shows that SAM-Semantics generates five candidate regions when directly applied to the original image. Identifying which regions correspond to the lesion is crucial, necessitating system intervention. In contrast, the objective of this study was to achieve fully automatic lesion detection without any manual intervention. To this end, YOLOv8 architecture was employed, and bounding box coordinates were utilized as inputs for SAM, facilitating the automatic detection of the lesion region.

The image given in Figure 8 shows a lesion located near the eye, sourced from a private database. Components such as the iris and eyelashes were included in acquiring the lesion image. In such cases, segmentation methods operating on the entire image may misinterpret non-lesion regions as lesions, potentially reducing their performance. This study deliberately incorporates images of clinical cases, like those illustrated, into the database. This inclusion aims to ensure that performance evaluation results are reliable for real-world applications, addressing situations commonly encountered in clinical practice and enhancing the robustness of the proposed method for practical implementations.

Table 1 was then compiled to show the entire system’s performance for four cases. The first and second cases involve training on the public database, with subsequent testing conducted on both public and private databases. Conversely, the third and fourth cases entail training on a hybrid database consisting of public and private images, followed by testing on private and hybrid image databases. This approach enables a comprehensive evaluation of segmentation performance, capturing a spectrum of conditions encountered across public and private image databases. Conducting a study focused on skin lesion segmentation, a private database is being curated alongside publicly available ones for training. Utilizing both public and private databases offers several advantages. Public data aids in generalizing the model, enabling it to comprehend diverse scenarios effectively.

Conversely, the private database focuses on specific application domains, enhancing the model’s performance within those contexts. Employing a mixed training approach that integrates both datasets improves the model’s adaptability to general and domain-specific data. This approach promotes diversity in training and bolsters the reliability of the model’s predictions.

Table 1 illustrates the performance evaluation of the YOLOv8 in detecting lesions and predicting bounding boxes. The assessment distinguishes between training these algorithms on a public or mixed database. As can be seen from the table, YOLOv8 exhibits robust performance across both training approaches. When trained on the public database, YOLOv8 achieves a test performance of 98.30% mAP, 96.20% precision, and 94.40% recall. Training on the mixed database results in slightly adjusted scores: 98% mAP, 96.90% precision, and 94.30% recall. The results underscore the efficacy of employing a mixed database strategy, which enhances the method’s performance to generalize across diverse lesion detection tasks. The findings highlight improvements in precision and recall metrics, emphasizing the significance of dataset diversity in optimizing algorithmic performance for practical applications in medical imaging.

Following the evaluation of YOLOv8, SAM-Box has been assessed across four different scenarios, with the highest scores highlighted in bold in the table. A comprehensive analysis using five metrics demonstrates the effectiveness of the proposed method in accurately determining skin lesion regions. This underscores a significant advancement towards developing a fully automated reporting (decision support) system, achieved by integrating a classification step after the segmentation phase to achieve optimal performance. According to the results, the highest scores for YOLOv8 and SAM-Box in Case 1 are as follows: Dice coefficient, Jaccard index, accuracy, recall, and precision, with values of 0.9399, 0.9112, 0.9589, 0.9308, and 0.9793, respectively.

The observation regarding the performance decrease in Case 3 (training on the hybrid database) compared to Case 1 (training on the public database) can be affected by several factors explained below:Variability in Image Quality: Private clinical databases often come from heterogeneous sources, leading to significant variations in image quality, lighting conditions, and resolution compared to standardized public databases. These discrepancies can hinder the model’s ability to generalize effectively.Lesion Size and Characteristics: The private database contains lesions with varying sizes, shapes, or appearances that differ from those in the public database, potentially causing lower performance metrics. Additionally, the presence of rare lesion types or atypical characteristics in the private database might not be adequately represented in the public dataset, leading to reduced generalization.Skin Tone Variation: The private database includes a more diverse range of skin tones, which can affect image contrast and model performance. Public databases typically have limited representation of skin tones, which can influence the model’s robustness across different populations.

These factors contribute to decreased Dice and Jaccard scores when transitioning from the public to the hybrid database. Regarding the model’s generalizability, an external validation on independent clinical databases would be an important step in further assessing the robustness of the YOLOv8 and SAM-Box system. This would help evaluate how well the model performs across different clinical environments and patient demographics, enhancing its applicability and robustness. Therefore, our efforts to collect clinical data from diverse sources are ongoing.

In addition to the performance evaluation of YOLOv8, the latest YOLO: YOLOv11 is considered in the proposed study. A detailed comparison between YOLOv8 and YOLOv11 is performed using the constructed database, and the results support the choice of YOLOv8. The YOLOv11 performance for the skin lesion segmentation task is given in Table 2. Regarding detection performance, YOLOv8 demonstrated superior or comparable results across all metrics. If training was performed on a public database (Cases 1 and 2), YOLOv8 achieved an mAP50 of 98.3%, precision of 96.2%, and recall of 94.4%, while YOLOv11 showed an mAP50 of 98.3%, precision of 95.8%, and recall of 95.4%. These results were consistent across additional cases, highlighting YOLOv8’s high accuracy and reliability.

Regarding segmentation tasks, YOLOv8 outperformed YOLOv11 by a significant margin. For example, in Case 1, YOLOv8 achieved a Dice score of 0.9399 and a Jaccard score of 0.9112, while YOLOv11 provided 0.8394 and 0.7433, respectively. So, the YOLOv8 provides more precise and robust segmentation, which is crucial for the goals of our study.

Furthermore, the performance of YOLOv8 and YOLOv11 is compared to show the method’s computational efficiency and memory requirements. The simulations used an MSI laptop with 32 GB RAM and an NVIDIA RTX 4070 8 GB GPU (CUDA). The results are given in Table 3. Regarding computational efficiency, YOLOv8 demonstrated faster inference times, with an average of 6.99 ms per prediction, compared to YOLOv11’s 10.42 ms. Lower inference time makes YOLOv8 more suitable for real-time applications if speed is essential. Although YOLOv8 has more parameters (43M) and higher GFLOPs (165.4) than YOLOv11 (25M parameters and 87.3 GFLOPs), the segmentation and total elapsed times were similar: 1.05 s for YOLOv8 versus 1.06 s for YOLOv11, with total times of 164.57 s for YOLOv8 and 163.57 s for YOLOv11. Thus, YOLOv8 efficiently uses computational resources without compromising performance. Based on these findings, YOLOv8 is preferred for our study due to its consistently superior performance in detection and segmentation, faster inference times, and proven reliability in the research community. While YOLOv8 is currently optimal for the given work, we acknowledge the rapid advancements in object detection models and plan to explore newer versions of YOLO and other state-of-the-art models in future research to enhance the performance of our framework further.

Finally, the observation regarding handling challenging cases, such as lesions obscured by hair or those with complex backgrounds, is considered. In such edge cases, the proposed YOLOSAMIC system utilizes the bounding box generated by YOLOv8 (Large) to guide the SAM-Box segmentation process. This pre-localization step significantly improves the model’s ability to distinguish lesions from non-lesion regions, especially in cases where lesion boundaries are ambiguous. However, these edge cases still present inherent challenges, such as the possibility of false positives from non-lesion regions or difficulties in detecting complex lesion boundaries.

So, compared to typical cases, a detailed analysis was performed by computing separate metrics for difficult cases (e.g., lesions obscured by hair, small lesions, and those with complex backgrounds) to better evaluate the system’s reliability in such scenarios. These metrics, including the Dice coefficient, Jaccard index, accuracy, recall, precision, and specificity, were calculated explicitly for complex cases, enabling a clear performance comparison. This additional analysis offers more profound insights into the system’s robustness and applicability to real-world clinical environments, particularly for cases that may not conform to the idealized conditions seen in typical datasets. Table 4 provides the segmentation results for difficult and typical cases.

Based on the table, the YOLOv8 and SAM-Box-based segmentation approach shows lower performance in previously noted challenging cases than typical ones, which is an expected outcome. A closer look at the table reveals that the decline is more pronounced in Dice and Jaccard scores, indicating a reduced overlap between the ground truth and predicted masks. However, metrics such as accuracy, precision, and specificity derived from the pixel-based confusion matrix do not exhibit a significant decrease. These results suggest that while the method struggles with precise boundary detection in complex cases, its overall classification ability remains relatively stable.

Our study significantly advances skin cancer diagnosis by integrating cutting-edge algorithms. These technologies offer state-of-the-art real-time efficiency and accuracy, outperforming the earlier models. The performance metrics achieved in the presented research are particularly noteworthy. The obtained mAP, precision, and recall values are about 98%, 96%, and 94%. These results indicate high accuracy and reliability in object detection, significantly improving over other models. Compared to existing studies in Table 5, the proposed system excels in balancing accuracy, automation, and clinical applicability. The proposed study integrates a hybrid dataset combining public and private clinical images. This inclusion is pivotal, as it rigorously tests the system against real-world challenges, such as lesions obscured by hair or complex backgrounds. The Dice and Jaccard indices of 0.9399 and 0.9112 on public datasets highlight the system’s superior segmentation precision, surpassing most prior studies. Furthermore, YOLOv8 exhibited higher specificity in all cases. According to the results, YOLOv8 is more effective at minimizing false positives and maximizing true positives, which is vital for our application.

Table 5 presents a comparative analysis of our method with existing state-of-the-art segmentation models. While not all compared methods were tested on the same dataset, a fair evaluation is ensured by using ISIC 2016–2018 images, which are commonly employed in prior works. Due to ethical constraints, it is impossible to retrain existing SOTA methods on our hybrid dataset, as this would require access to models and training details that are not publicly available. The results highlight the effectiveness of our approach in handling real-world clinical challenges, reinforcing its robustness and generalizability.

Furthermore, our hybrid dataset incorporates public and private clinical images, ensuring a more realistic evaluation that accounts for challenges such as lesion size variability, skin tone diversity, and complex backgrounds. Due to ethical constraints, it is impossible to publicly share our private clinical dataset, as it contains sensitive patient data collected under an institutional-approved protocol. Additionally, re-implementing existing SOTA methods on our hybrid dataset is not feasible, as this would require access to the original models, including source code, hyperparameters, and training pipelines, many of which are unavailable or not reproducible based on published works. Despite this limitation, our results demonstrate that integrating both public and private data improves segmentation performance, underscoring the clinical applicability of the proposed approach.

While some methods achieve high accuracy through manual intervention or task-specific fine-tuning, our approach combines YOLOv8’s advanced object detection with SAM-Box’s segmentation capabilities to deliver a fully automated workflow. This innovation streamlines the diagnostic process and demonstrates exceptional adaptability to diverse clinical conditions. This study sets a new standard in automation and reliability, paving the way for practical deployment in dermatology practices.

### 3.3. Ablation Study

This ablation study aims to evaluate the performance of YOLOv8 under various conditions to understand the impact of different model components and settings on their overall effectiveness. Ablation analysis focuses on the ability of YOLOv8 under specific image processing scenarios, including grayscale images, blurred images, and pruned models.

YOLOv8’s performance on grayscale images is critical for evaluating how well the model handles reduced color information. The results show that YOLOv8 achieves a Dice score of 0.7938 and a Jaccard index of 0.7012, demonstrating robust segmentation ability even when working with monochromatic images (Table 6). Simulations show that YOLOv8 can effectively differentiate structures in grayscale images, making it suitable for limited color information. In real-world applications, images may suffer from blurring, which can negatively impact segmentation performance. So, YOLOv8 is performed on blurred images to simulate this condition. As shown in Table 6, YOLOv8 achieved a Dice score of 0.8666 and a Jaccard index of 0.7791. These values indicate YOLOv8’s strong ability to perform segmentation even under blurred conditions.

Additionally, YOLOv8 maintained a high specificity, suggesting that the model minimizes false positives in challenging conditions. Finally, to evaluate YOLOv8’s robustness when its model size is reduced, various tests are carried out at a 10% pruning level. The results presented in Table 6 show that YOLOv8 continues to perform well even with pruning, although there is a drop in performance as the pruning level increases. For the 10% pruned model, YOLOv8 achieved a Dice score of 0.8629 and a Jaccard index of 0.77398, demonstrating that the model retains its segmentation ability with minor pruning.

The ablation study demonstrates the effectiveness of YOLOv8 across a variety of conditions. Even with reduced color information, blurred images, and pruned model configurations, YOLOv8 maintains strong segmentation performance. While pruning does impact the model’s performance, YOLOv8’s robustness to these changes emphasizes its potential for real-world applications, where both computational efficiency and segmentation accuracy are essential.

In the final part of the experiments, a small and hybrid database is constructed, including 500 images to provide a more detailed justification for the choice of YOLOv8 (Large) and SAM’s ViT-H variants. First, the bounding box detection performance of YOLOv8 configurations is investigated. Then, the experiments for SAM-Box variants were performed after YOLOv8 (Large) was carried out. The results achieved for YOLOv8 configurations are given in Table 7. A trade-off between detection accuracy and computational feasibility drove the choice of YOLOv8 (Large) over smaller variants. Given the critical importance of precise lesion localization in medical imaging, the large variant provides superior feature extraction capabilities, leading to higher detection performance (94.61% mAP, 93.75% precision, and 92.57% recall) compared to its smaller counterparts. YOLOv8’s transformer-based feature fusion enhances contextual understanding, which is crucial for detecting lesions with subtle differences in contrast.

SAM-Box versions were examined after deciding to use the YOLOv8 (Large) model. The results are given in Table 8. According to the experiments, ViT-H was selected for the SAM-Box segmentation stage as it demonstrated the best performance among all available SAM versions. While ViT-B and ViT-L are more computationally efficient, ViT-H offers higher segmentation fidelity, particularly in challenging cases with irregular lesion boundaries. Experimental evaluations confirmed that ViT-H significantly improved segmentation performance, achieving a Dice coefficient of 0.8875 and a Jaccard index of 0.8010 on the constructed database. Furthermore, confusion matrix-based metrics are considerably higher than the remaining versions. The selection of these configurations thus balances computational cost with segmentation accuracy, ensuring optimal performance for real-world clinical applications.

In conclusion, the findings of this study contribute to the field of medical image analysis by demonstrating how hybrid deep-learning architecture can enhance segmentation performance, particularly in medical imaging applications. Theoretically, integrating YOLOv8 with SAM-Box shows the potential of combining object detection with segmentation to improve lesion localization and delineation.

From a practical standpoint, the proposed system offers a scalable and efficient approach for automated skin lesion analysis, reducing the reliance on manual annotations and improving diagnostic accessibility. Including public and private datasets ensures generalizability, making this method viable for real-world clinical applications.

## 4. Conclusions

This study introduces an innovative and fully automated skin lesion segmentation system that synergizes YOLOv8 and SAM-Box for accurate and reliable performance. The proposed method addresses critical challenges in dermatology, including lesion appearance variability and expert dermatologists’ scarcity, by combining object detection and segmentation techniques. The system can operate across diverse datasets, including public and private clinical images, emphasizing its robustness and practical use.

The integration of YOLOv8 and SAM-Box achieved satisfactory precision, recall, Dice, and Jaccard indices, demonstrating substantial improvements over existing approaches. The hybrid dataset, including real-world conditions like hidden lesions and complicated backgrounds, proved the system is reliable and effective. These results emphasize the significant potential of this automated solution to improve diagnostic processes and aid in clinical decision-making.

Furthermore, the critical role of ablation experiments in evaluating the performance of YOLOv8 in medical image segmentation is highlighted. The ablation results emphasize the model’s ability to adapt to complex and noisy data. By comparing YOLOv8 with version v11, we have demonstrated that the enhancements in YOLOv8’s architecture led to notable improvements in segmentation accuracy and precision.

This study aims to improve segmentation methods and expand the dataset to include a wider variety of lesion types. Future clinical trials will be essential in evaluating how well the model works in real medical settings. Overall, this research represents an important step toward creating accessible, accurate, and scalable tools for early skin cancer detection, supporting global healthcare goals to enhance patient outcomes with the help of technological advancements.


## Figures and Tables

**Figure 1 diagnostics-15-00479-f001:**
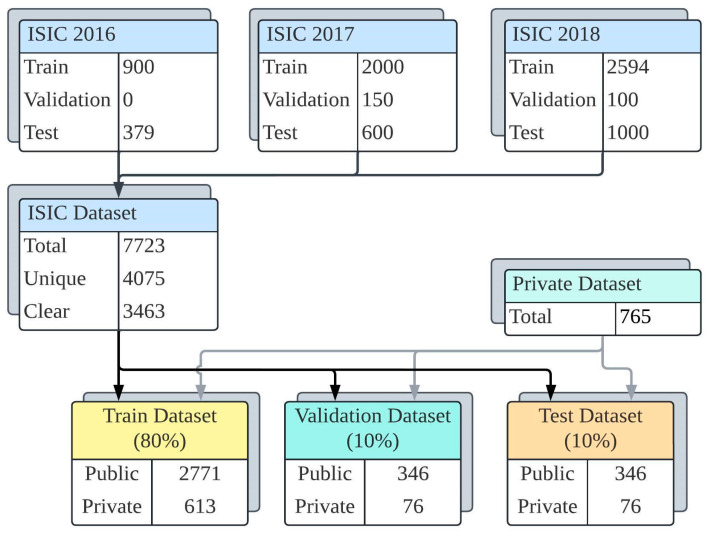
The database of the study.

**Figure 2 diagnostics-15-00479-f002:**
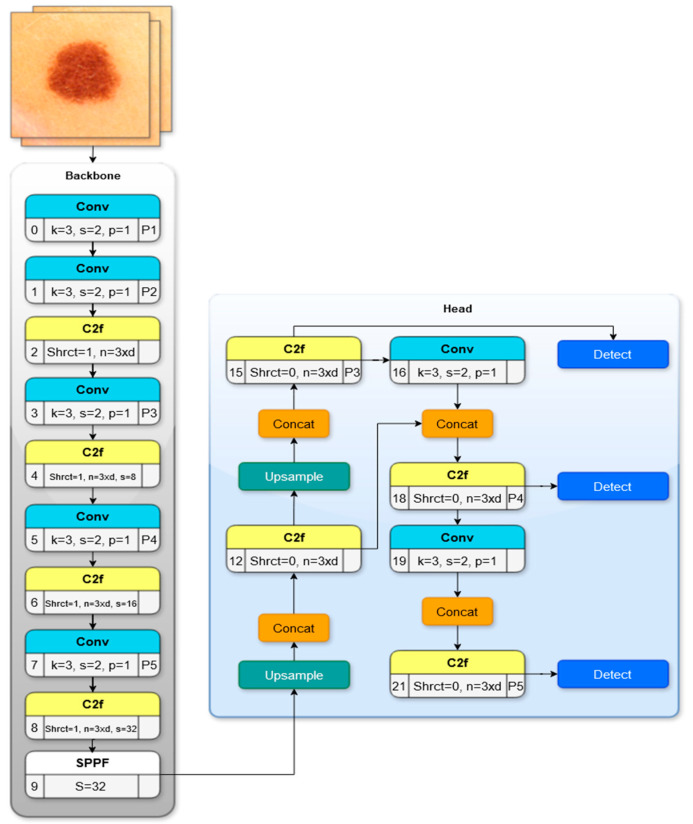
YOLOv8 architecture.

**Figure 3 diagnostics-15-00479-f003:**
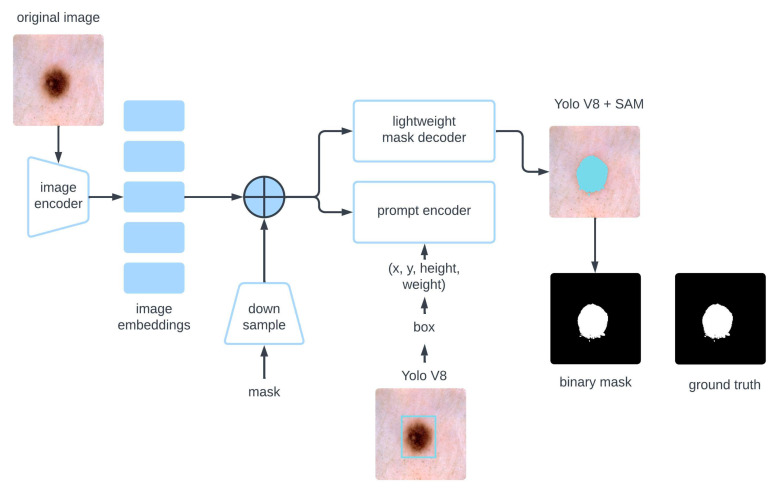
SAM-based scheme of the proposed study.

**Figure 4 diagnostics-15-00479-f004:**
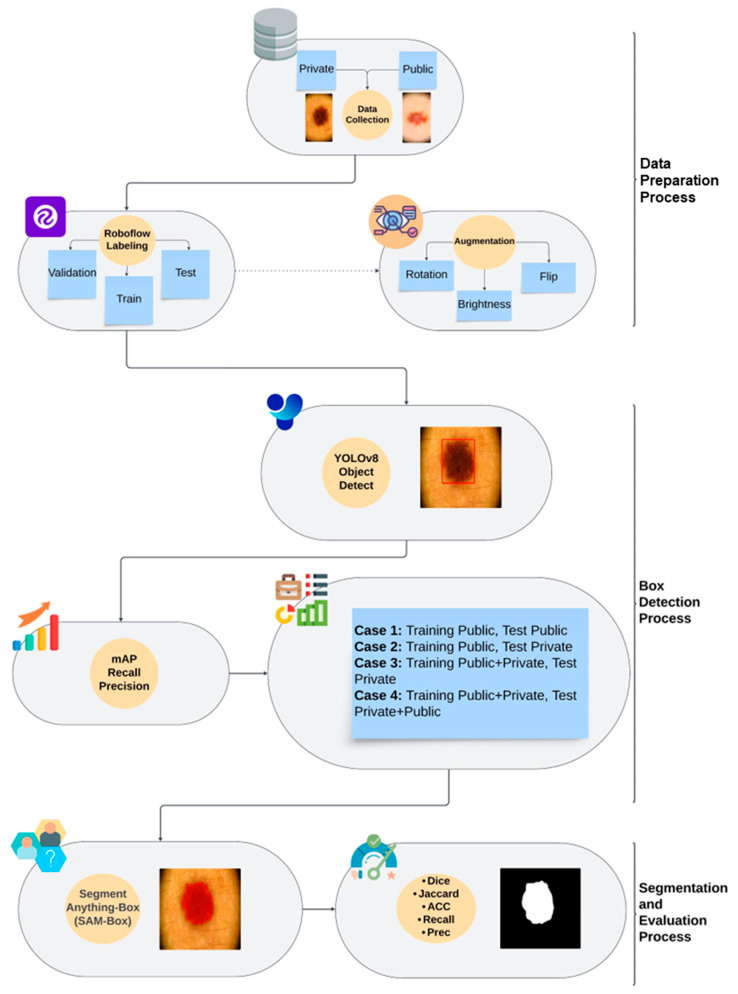
The flowchart of the study.

**Figure 5 diagnostics-15-00479-f005:**
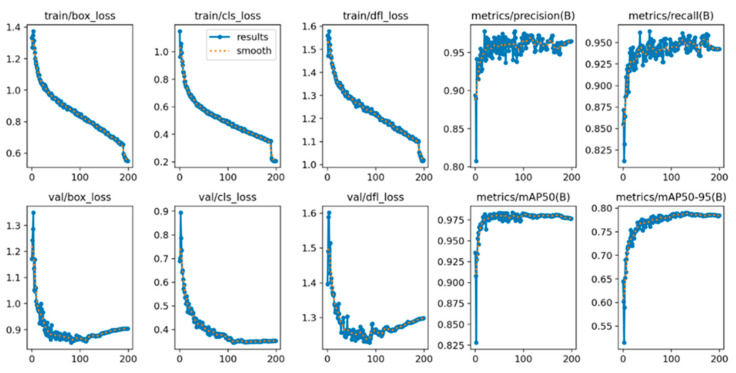
The training performance of YOLOv8 on the public database.

**Figure 6 diagnostics-15-00479-f006:**
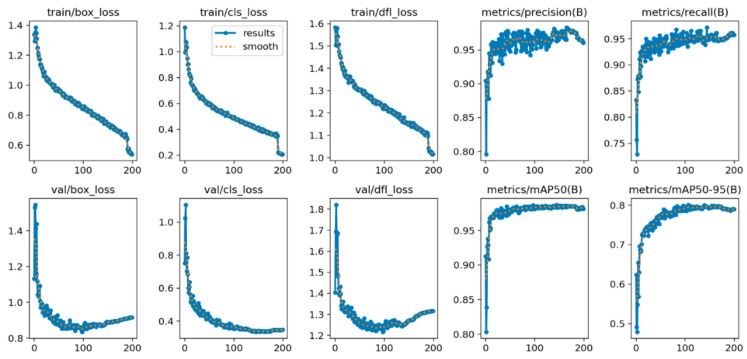
The training performance of YOLOv8 on the hybrid database.

**Figure 7 diagnostics-15-00479-f007:**
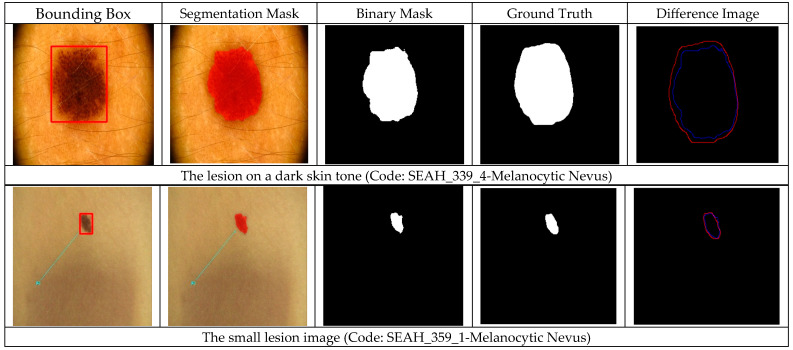
Segmentation results of challenging cases. (Blue and red lines show the predicted and ground-truth lesion boundaries, respectively).

**Figure 8 diagnostics-15-00479-f008:**
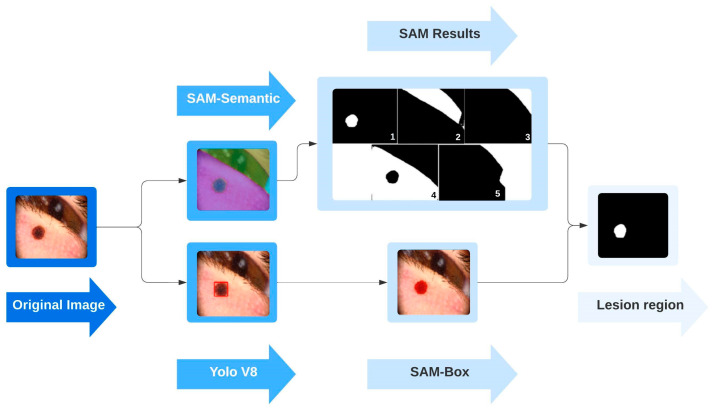
SAM outputs for a challenging skin lesion image.

**Table 1 diagnostics-15-00479-t001:** The result of the proposed skin lesion segmentation system for four training/test scenarios.

		Database		YOLOv8 Test Results (%)			SAM-Box Results				
		Public	Private	mAP50	Precision	Recall	Dice	Jaccard	ACC	Recall	Precision
Case 1	**Training**	✓		98.30	96.20	94.40	**0.9399**	**0.9112**	**0.9589**	**0.9308**	**0.9793**
	**Test**	✓									
Case 2	**Training**	✓					0.8901	0.8129	0.9569	0.9595	0.8424
	**Test**		✓								
Case 3	**Training**	✓	✓	98.00	96.90	94.30	0.8679	0.7887	0.9492	0.9381	0.8392
	**Test**		✓								
Case 4	**Training**	✓	✓				0.8990	0.8445	0.9552	0.9282	0.9033
	**Test**	✓	✓								

**Table 2 diagnostics-15-00479-t002:** The result of the YOLOv11-based lesion segmentation for four training/test scenarios.

		Database		YOLOv11 Test Results (%)			SAM-Box Results				
		Public	Private	mAP50	Precision	Recall	Dice	Jaccard	ACC	Recall	Precision
Case 1	**Training**	✓		98.3	95.8	95.4	0.8394	0.8433	0.9296	0.8371	0.8827
	**Test**	✓									
Case 2	**Training**	✓					0.8764	0.7975	0.9443	0.8811	0.8994
	**Test**		✓								
Case 3	**Training**	✓	✓	98.3	95.8	94	0.8641	0.7874	0.9507	0.8399	0.9117
	**Test**		✓								
Case 4	**Training**	✓	✓				0.8588	0.7729	0.9442	0.8274	0.9190
	**Test**	✓	✓								

**Table 3 diagnostics-15-00479-t003:** Computational complexity comparison.

	YOLOv8	YOLOv11
Layers	365	631
Parameters	43.630.611	25.311.251
Gradients	43.630.595	25.311.235
GFLOPs	165.4	87.3
Avarage Segmentation Time (s)	1.05	1.05
Average Prediction Time (ms)	9.88	11.77
Total Time Elapsed (s)	164.57	163.57

**Table 4 diagnostics-15-00479-t004:** Segmentation performance for difficult and typical cases.

Typical Cases						
Model	Dice	Jaccard	ACC	Recall	Precision	Specificity
YOLOv8	0.8782	0.7924	0.9604	0.8332	0.9466	0.9816
Difficult Cases						
Model	Dice	Jaccard	ACC	Recall	Precision	Specificity
YOLOv8	0.8198	0.7254	0.8797	0.8292	0.8810	0.8676

**Table 5 diagnostics-15-00479-t005:** Comparison of the proposed system with recent studies.

Reference No.	Data	Model	Dice	Jaccard	Accuracy	Recall	Precision	Specificity
[9]	ISIC 2018 PH2 DermIS	U-net + LinkNet	0.8940	0.8090	0.9690	-	-	
[10]	ISIC	FSPBO-DQN	-	-	0.9236	-	-	
[11]	ISIC 2016 ISIC 2017	FCEDN	0.9848 0.8723	0.9641 0.8685	0.9832 0.9525	-	-	
[12]	HAM10000	SkinSAM	0.8879	0.7843	0.9450	-	-	
[13]	ISIC PH2	LinkNet-B7	0.9675 0.9742	-	-	-	-	
[14]	ISIC 2017 ISIC 2018 PH2	(MSAU-Net)	0.9032 0.8980 0.9377	0.9576 0.9560 0.9617	0.9576 0.9490 0.9617	-	-	
[15]	ISIC	MD^2^N	0.9718	-	0.9746	0.9972	0.9363	
[32]	HAM10000	S^2^C-DeLeNet	0.9467	0.9007	0.9720	0.9373	0.9593	
Ours	Public	YOLOv8 + SAM-Box	0.9399	0.9112	0.9589	0.9308	0.9793	0.9739
	Hybrid	YOLOv8 + SAM-Box	0.8990	0.8445	0.9552	0.9282	0.9033	0.9735

**Table 6 diagnostics-15-00479-t006:** Ablation results.

YOLOv8						
	Dice	Jaccard	ACC	Recall	Precision	Specificity
Gray Image	0.7938	0.7012	0.8547	0.8363	0.8431	0.8370
Blurred Image	0.8666	0.7791	0.9444	0.8325	0.9336	0.9590
%10 Pruned	0.8629	0.7740	0.9440	0.8250	0.9359	0.9592

**Table 7 diagnostics-15-00479-t007:** Performance evaluation of YOLOv8 configurations.

Bounding-Box Results (%)			
Model	mAP50	Precision	Recall
**YOLOv8n**	94.58	93.41	90.24
**YOLOv8s**	93.88	94.4	87.76
**YOLOv8m**	93.67	95.03	82.95
**YOLOv8l**	**94.61**	**93.75**	**92.57**
**YOLOv8x**	91.51	94.13	83.44

**Table 8 diagnostics-15-00479-t008:** Segmentation performance of SAM-Box ViT versions.

SAM-Box Results with YOLOv8 Large Model						
	Dice	Jaccard	ACC	Recall	Precision	Specificity
**SAM-ViT-B**	0.8801	0.8012	0.9442	0.8819	0.9091	0.9562
**SAM-ViT-L**	0.8770	0.7953	0.9486	0.8476	0.9388	0.9736
**SAM-ViT-H**	**0.8875**	**0.8010**	**0.9531**	**0.8883**	**0.9256**	**0.9793**

## Data Availability

Data are contained within the article but cannot be shared due to ethical committee decision.

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
