# Peer review of "YOLOSAMIC: A Hybrid Approach to Skin Cancer Segmentation with the Segment Anything Model and YOLOv8"

_diagnostics, 2025, doi:10.3390/diagnostics15040479_

Round 1
Reviewer 1 Report
Comments and Suggestions for Authors
Dear Authors,
Thank you so much for submitting this valuable manuscript. The paper is well-presented and organized.
1. However, I have a serious concern about why you have considered the YoloV8 while the latest versions are also available publicly.
2. Contributions points need to be refined.
3. There are some typo mistakes, like in Table 2, where "," instead of "." is in the accuracy column. Please check thoroughly.
4. Add theoretical and practical implications of the study before conclusion.
5. Add ablation study.
Author Response
The author's reply is attached as a Word file. Sincerely...

Reviewer 2 Report
Comments and Suggestions for Authors
YOLOSAMIC: A Hybrid Approach to Skin Cancer Segmentation with Segment Anything Model and YOLOv8
My comments and suggestions are the following:
1. The authors demonstrate robust performance for both public and hybrid datasets in Table 1, highlighting the strength of the YOLOv8 and SAM-Box system. However, in Case 3 (training on a hybrid dataset), the Dice coefficient and Jaccard index are notably lower compared to Case 1 (training on a public dataset). What factors could contribute to the decrease in performance when incorporating private clinical images? Additionally, can the authors provide more details on the characteristics of the private dataset (e.g., variations in lesion size, skin tone, or image quality) and its influence on generalizability? Would external validation on other independent clinical datasets further enhance the system's robustness?
2. Figure 7 showcases segmentation results for challenging cases, such as lesions obscured by hair or those with complex backgrounds. While the segmentation results are visually promising, how does the system handle edge cases where the lesion boundaries are highly ambiguous or overlap with non-lesion regions? Additionally, could the authors quantify and compare segmentation performance specifically for these challenging scenarios (e.g., by computing separate metrics for difficult cases versus typical cases)? Such analysis would provide deeper insights into the reliability of the proposed system in real-world settings.
3. I suggest that the authors consider incorporating a discussion of relevant studies in the 'Introduction' section, which could further enrich the context of their research. For instance, the studies “Automated multi-class classification of skin lesions through deep convolutional neural network with dermoscopic images” and “Deep learning‐based automated detection of human knee joint's synovial fluid from magnetic resonance images with transfer learning” provide recent insights into automated skin cancer analysis and novel classification and segmentation methods using machine learning models. Including such related works could offer valuable perspectives for the current manuscript, and similar studies in this field may also be highlighted to underscore advancements in related methodologies.
4. To enhance the manuscript's presentation, I suggest a comprehensive proofreading and editing process. This should focus on improving clarity, grammatical correctness, and overall sentence structure, which would significantly enhance readability. Attention to grammar, punctuation, and language consistency would also be beneficial.
5. The manuscript introduces YOLOSAMIC, which integrates YOLOv8 for object detection with SAM-Box for segmentation. While this combination utilizes state-of-the-art models, SAM-based segmentation methods have previously shown limitations in medical imaging due to low contrast and structural complexity. How does the proposed system address these limitations to achieve higher Dice and Jaccard indices compared to standalone SAM or other segmentation models (e.g., U-Net, LinkNet)? Furthermore, can the authors provide a more detailed justification for the choice of YOLOv8 (Large) and ViT-H variants over smaller or alternative configurations?
6. In Table 2, the authors compare their system's performance with prior studies using various segmentation models. While YOLOSAMIC achieves higher Dice and Jaccard indices for public datasets, the metrics for hybrid datasets are slightly lower. How does the system's performance compare when evaluated specifically on clinically challenging conditions, such as small lesions or lesions in individuals with darker skin tones, as highlighted in the manuscript? Would the inclusion of additional evaluation metrics, such as False Positive Rate (FPR) or sensitivity to clinical variations, provide a more comprehensive comparison with existing methods?
Comments on the Quality of English Language
To enhance the manuscript's presentation, I suggest a comprehensive proofreading and editing process. This should focus on improving clarity, grammatical correctness, and overall sentence structure, which would significantly enhance readability. Attention to grammar, punctuation, and language consistency would also be beneficial.
Author Response

(The authors gave the same response as above.)

Reviewer 3 Report
Comments and Suggestions for Authors
Author have presented A Hybrid Approach to Skin Cancer Segmenta-2 tion with Segment Anything Model and YOLOv8.
Here are few observations.
SAM model is already is already have 1 million parameters. Method is becoming overcomplicated by adding Yolo Network that too Yolo 8.
What specific limitations of SAM does YOLOv8 address?
How much this complicated structure is feasible for health applications.
It is expected to evaluate method’s computational efficiency and memory requirements.
Can you provide comparison in terms of inference time and hardware dependency?
Figure 4 is not clear and too general; SAM model is not reflected in Figure 4.
Please show difference image of predicted output and ground truth image in Figure 7.
Figure 8 citation is coming before Figure 7. Please check. Line 472 and line 479.
Comparative analysis shown in Table 2 is not proper. Need to show SOTA on the same image dataset.
Also first two contribution claimed by author are too obvious.
Author Response

(The authors gave the same response as above.)

Round 2
Reviewer 2 Report
Comments and Suggestions for Authors
Dear Editor,
Thank you for the opportunity to review this manuscript. I have carefully examined the revised version along with the authors' responses to the previous comments.
I am pleased to recommend this manuscript for publication in your journal, as the authors have effectively addressed my suggestions, resulting in significant improvements to the clarity, methodology, and overall scientific rigor of the study.
Furthermore, this research holds substantial significance within the scientific community, particularly in the domain of Skin Cancer Segmentation. The proposed approach presents methodological advancements that can contribute to ongoing research and practical applications, making it a valuable addition to the literature.
I appreciate the opportunity to contribute to the peer review process and look forward to seeing this work published.
Thank you
Author Response
Comments:
Thank you for the opportunity to review this manuscript. I have carefully examined the revised version along with the authors' responses to the previous comments. I am pleased to recommend this manuscript for publication in your journal, as the authors have effectively addressed my suggestions, resulting in significant improvements to the clarity, methodology, and overall scientific rigor of the study. Furthermore, this research holds substantial significance within the scientific community, particularly in the domain of Skin Cancer Segmentation. The proposed approach presents methodological advancements that can contribute to ongoing research and practical applications, making it a valuable addition to the literature. I appreciate the opportunity to contribute to the peer review process and look forward to seeing this work published.
Response: Dear Reviewer;
We sincerely appreciate your valuable contributions. Your insightful comments have greatly enhanced the quality of the paper.
Reviewer 3 Report
Comments and Suggestions for Authors
Author have addressed the raised comments. Paper may be accepted subject to improvement in quality of images.
Author Response
Comment: Author have addressed the raised comments. Paper may be accepted subject to improvement in quality of images.
Response: Dear Reviewer;
We sincerely appreciate your feedback. All the raised comments have been carefully addressed, and we will enhance the image quality as suggested. Thank you for your time and consideration.